# DYRK1A and Activity-Dependent Neuroprotective Protein Comparative Diagnosis Interest in Cerebrospinal Fluid and Plasma in the Context of Alzheimer-Related Cognitive Impairment in Down Syndrome Patients

**DOI:** 10.3390/biomedicines10061380

**Published:** 2022-06-10

**Authors:** Manon Moreau, Maria Carmona-Iragui, Miren Altuna, Lorraine Dalzon, Isabel Barroeta, Marie Vilaire, Sophie Durand, Juan Fortea, Anne-Sophie Rebillat, Nathalie Janel

**Affiliations:** 1CNRS, UMR 8251, Biologie Fonctionnelle et Adaptative (BFA), Université Paris Cité, 75013 Paris, France; manon.moreau@live.fr (M.M.); lorraine.dalzon@gmail.com (L.D.); 2Sant Pau Memory Unit, Department of Neurology, Hospital de la Santa Creu i Sant Pau, Biomedical Research Institute Sant Pau, Universitat Autònoma de Barcelona, 08193 Barcelona, Spain; mcarmonai@santpau.cat (M.C.-I.); maltuna@santpau.cat (M.A.); ibarroeta@santpau.cat (I.B.); jfortea@santpau.cat (J.F.); 3Center of Biomedical Investigation Network for Neurodegenerative Diseases (CIBERNED), 28029 Madrid, Spain; 4Barcelona Down Medical Center, Fundació Catalana de Síndrome de Down, 08029 Barcelona, Spain; 5Institut Médical Jérôme Lejeune, 75015 Paris, France; marie.vilaire@institutlejeune.org (M.V.); sophie.durand@institutlejeune.org (S.D.); annesophie.rebillat@institutlejeune.org (A.-S.R.)

**Keywords:** Down syndrome, Alzheimer’s disease, DYRK1A, ADNP, brain, plasma, CSF

## Abstract

Down syndrome (DS) is a complex genetic condition due to an additional copy of human chromosome 21, which results in the deregulation of many genes. In addition to the intellectual disability associated with DS, adults with DS also have an ultrahigh risk of developing early onset Alzheimer’s disease dementia. DYRK1A, a proline-directed serine/threonine kinase, whose gene is located on chromosome 21, has recently emerged as a promising plasma biomarker in patients with sporadic Alzheimer’s disease (AD). The protein DYRK1A is truncated in symptomatic AD, the increased truncated form being associated with a decrease in the level of full-length form. Activity-dependent neuroprotective protein (ADNP), a key protein for the brain development, has been demonstrated to be a useful marker for symptomatic AD and disease progression. In this study, we evaluated DYRK1A and ADNP in CSF and plasma of adults with DS and explored the relationship between these proteins. We used mice models to evaluate the effect of DYRK1A overexpression on ADNP levels and then performed a dual-center cross-sectional human study in adults with DS in Barcelona (Spain) and Paris (France). Both cohorts included adults with DS at different stages of the continuum of AD: asymptomatic AD (aDS), prodromal AD (pDS), and AD dementia (dDS). Non-trisomic controls and patients with sporadic AD dementia were included for comparison. Full-form levels of DYRK1A were decreased in plasma and CSF in adults with DS and symptomatic AD (pDS and dDS) compared to aDS, and in patients with sporadic AD compared to controls. On the contrary, the truncated form of DYRK1A was found to increase both in CSF and plasma in adults with DS and symptomatic AD and in patients with sporadic AD with respect to aDS and controls. ADNP levels showed a more complex structure. ADNP levels increased in aDS groups vs. controls, in agreement with the increase in levels found in the brains of mice overexpressing DYRK1A. However, symptomatic individuals had lower levels than aDS individuals. Our results show that the comparison between full-length and truncated-form levels of DYRK1A coupled with ADNP levels could be used in trials targeting pathophysiological mechanisms of dementia in individuals with DS.

## 1. Introduction

Down syndrome (DS, or trisomy 21), the most common chromosomal abnormality in humans, is a complex genetic condition due to an extra copy of human chromosome 21 (HSA21), which results in the deregulation of many genes [1]. DS is associated with various clinical conditions [1], and all individuals with DS have some degree of intellectual disability [2]. DS people have an increased risk of developing early onset Alzheimer’s disease (AD) dementia [3]. All people with DS at 40 years have Alzheimer’s-like neuropathological lesions, which disrupt brain function and increase the risk for symptomatic AD. People with DS have the neuropathological hallmarks of AD, with a progressive build-up of extracellular Aβ plaques and intraneuronal hyperphosphorylated tau [4,5].

As in sporadic AD, in DS, there is an AD continuum that encompasses a preclinical stage (asymptomatic DS patients (DS)), a prodromal stage (with cognitive impairment and behavioral changes but before dementia (DS/PROD), and a dementia stage of AD (DS/AD). DS brains exhibit Aβ diffuse plaques starting at age 12–13 years, which are observed in the preclinical and prodromal stages of sporadic AD [6,7].

This high prevalence in AD is thought to be due to the additional copy of the Amyloid Precursor Protein (*APP*) gene, coded on HSA21 [8]. An extra copy of the *APP* gene increases predisposition to early onset AD in DS people by enhancing the levels of amyloid-β (Aβ) [6]. DS individuals have a higher incidence of dementia compared to the overall population, the incidence of dementia being higher in individuals with DS than in the general population, with a progressive cognitive damage developing at a much earlier age, the median age of dementia onset across all reported studies being 55 years [9].

Numerous studies focused on molecular mechanisms that promote AD neuropathology in DS [10]. Trisomy of HSA21 results in increased gene dose for many other genes beyond *APP* playing a crucial part in DS neuropathology. Among these genes, *DYRK1A* (for Dual specificity Tyrosine (Y) phosphorylation-Regulated Kinase 1A) plays a well-known part in abnormal tau protein phosphorylation, a crucial mechanism underlying the formation of toxic neurofibrillary tangles (NFTs) in AD [11].

Biomarkers play a critical role in the characterization of the progression of a disease, and in both target engagement and theragnostic endpoints in drug development. Core biomarkers of AD have been studied in adults with DS, namely cerebrospinal fluid (CSF) Aβ1–42 and amyloid PET, CSF phosphorylated tau and tau PET, neuronal injury biomarkers such as CSF total tau and neurofilament light chain (NfL), or regional brain atrophy on MRI [8,12]. These studies investigating imaging and fluid biomarkers have demonstrated a strikingly similar sequence of events or natural history to that described in sporadic AD and autosomal dominant AD [12]. Despite these similarities, there might be some differences in Aβ metabolism, neuronal injury, or neuroinflammation [13]. We have previously shown that protein levels of DYRK1A, a proline-directed serine/threonine kinase, whose gene is located on HSA21, are lower in plasma from sporadic AD patients compared to controls of the same age. This decrease was already observed in patients with mild cognitive impairment (MCI) [14,15]. We also detected lower DYRK1A protein levels in lymphoblastoid cell lines (LCLs) from patients with AD dementia [14]. DYRK1A is an important factor that contributes to intellectual disability, memory deficit, and AD-type dementia, the main characteristics of the DS phenotype [16,17]. The key functions of DYRK1A have been associated with the neurodevelopmental abnormalities observed in people with DS [18]. Recent studies also indicate that this kinase plays an important role in early neurodegeneration, neuronal loss, and dementia in AD [19]. The phenotypic follow-up of cases of neurodevelopmental disorders has allowed to explore specific clinical phenotypes in a genotype-first manner, such as in the case of genes such as *DYRK1A* and activity-dependent neuroprotective protein (*ADNP*) [20]. Alterations in these genes are associated with more severe neurodevelopmental delays [20]. ADNP is essential for brain formation and is linked to cognitive functions. ADNP involvement has been demonstrated in AD dementia, suggesting that altered ADNP blood levels [21] could potentially be used as a marker for AD progression.

Based on the data mentioned above, the involvement of DYRK1A and ADNP in neurodevelopmental disorders and AD pathology makes them good candidate biomarkers for AD in DS. To test this hypothesis, we first have evaluated the effect of DYRK1A overexpression on ADNP in DS mouse models with different genetic complexity [22,23], and have analyzed CSF and blood levels of DYRK1A and ADNP using cohorts of cognitively normal euploid controls, patients with sporadic AD, and adults with DS across the whole AD continuum [8,12]. 

## 2. Materials and Methods

### 2.1. Experimental Mice

All procedures were performed in agreement with the ethical standards of French and European regulations (European Communities Council Directive, 86/609/EEC). Official authorization from the French Ministry of Agriculture was allowed to realize research and experiments on animals (number authorization 75–369). Mice were housed in a controlled environment with unlimited access to food and water on a 12-h light/dark cycle. Experimental protocol was endorsed by the institutional animal care and use committee of University (CEEA40). The mBACtgDyrk1A strain carries a murine bacterial artificial chromosome with on copy of *Dyrk1A* [22]. The mBACtgDyrk1A and Dp(16)1Yey mice were maintained on a C57Bl/6J background and genotyped as described [22,23]. Two-month-old male mice from the same litter were used, with the number and suffering of mice being reduced as much as possible.

### 2.2. Cohorts of Patients

Adults with DS were recruited in the Hospital de la Santa Creu I Sant Pau (Barcelona, Spain) and Institute Jérôme Lejeune (Paris, France). The Protocol number is IIBSP-DOW-2018-38. Participants or their legal representative gave written informed consent, as approved by the ethics committee of Hospital Sant following the standards for medical research in humans recommended by the Declaration of Helsinki and reported to the Ministry of Justice according to the Spanish laws for research in people with intellectual disabilities. For France, material was provided by the CRB BioJeL of the Institute Jérôme Lejeune according to the declaration and authorization from the Ministry of Research for the use, exportation, and export of human samples. Subjects were required to understand and accept the study procedures to give their informed consent. Genetic screening of trisomy 21 aneuploidy was performed as previously described confirming full trisomy of HSA 21 [8]. Participants were classified by physicians and neuropsychologists into asymptomatic AD (without AD-related cognitive impairment) (aDS), prodromal AD (with suspicion of AD but symptoms did not fulfil criteria of dementia) (pDS), AD dementia (fulfilling criteria of dementia) (dDS) according to previously published criteria [8]. Control subjects, without cognitive or neurological disorders and normal levels of core CSF biomarkers of AD, and sporadic AD patients were selected from the Sant Pau Initiative on Neurodegeneration (SPIN) cohort [8]. 

### 2.3. Cell Lines and Culture Conditions

LCLs, extensively used to analyze genotype/phenotype correlation, result from B lymphocytes of control individuals (2N) and non-related DS with (DS/AD) or without (DS) AD as previously mentioned [24,25]. Cultures of LCLs were done in Opti-MEM/GlutaMax, with 1% penicillin and streptomycin (10,000 U/mL) and 5% fetal bovine serum (from a unique batch) (Invitrogen, Cergy, France), at 37 °C with 5% CO_2_ in a humidified incubator.

### 2.4. Protein Extraction and Analysis

Total brain protein samples were prepared by homogenizing brains in 500 µL phosphate-buffered saline with a cocktail of protease inhibitors (1 mM Pefabloc SC, 5 μg/mL E64, and 2.5 μg/mL Leupeptin). LCLs were harvested by centrifugation, washed in PBS, followed by another centrifugation, and stored at −80 °C. Cell lysates were obtained from 5 × 10^6^ cell pellets treated with 300 μL of lysis buffer (Tris, 50 mM, pH8; NaCl, 150 mM; Igepal, 1% (Sigma-Aldrich, Saint-Quentin-Fallavier, France)), SDS, 0.1% containing protease inhibitors (1 mM Pefabloc SC, 5 μg/mL E64, and 2.5 μg/mL Leupeptin). Protein concentrations were detected with the Bio-Rad Protein Assay reagent (Bio-Rad, Hercules, CA, USA). Proteins were subjected to SDS electrophoresis on acrylamide gels under reducing conditions and transferred to Hybond-C Extra membrane (GE Healthcare Europe GmbH, Saclay, France). To assess and confirm the relative amounts of proteins, we used a slot blot method after testing the specificity of antibodies by Western blotting. Protein preparations were blotted on a Hybond-C Extra membrane (GE Healthcare Europe GmbH) using a Bio-Dot SF Microfiltration Apparatus (Bio-Rad). After transfer, membranes were saturated by incubation in 10% *w*/*v* nonfat milk powder in Tris-saline buffer (1.5 mM Tris base, pH 8; 5 mM NaCl; 0.1% Tween-20) and incubated overnight at 4 °C with an antibody directed ADNP (1/500, Santa Cruz Biotechnology, catalog # sc-393377, Dallas, TX, USA), DYRK1A (1/500, Abnova corporation, catalog # NP_001387, Taipei, Taiwan), DYRK1A (Sigma-Aldrich, catalog # D1694). Binding of the primary antibody was detected by incubation with horseradish peroxidase (HRP)-conjugated secondary antibody using the Western Blotting Luminol Reagent (Santa Cruz Biotechnology). β-actin (for Western blot) (Sigma-Aldrich) or Ponceau-S coloration (for slot blot) (Sigma-Aldrich) was used as an internal control. Digitized images of the immunoblots obtained using an LAS-3000 imaging system (Fuji Photo Film Co., Ltd., Tokyo, Japan) were used for densitometric measurements with an image analyzer (UnScan It software, Silk Scientific Inc., Orem, UT, USA). 

### 2.5. Preparation of Plasma and CSF Samples and Essays

Plasma and CSF samples were collected according to international consensus recommendations, processed, aliquoted, and frozen at −80 °C according to standardized procedures [8]. The DYRK1A levels in plasma and CSF were assessed by a solid phase immobilized epitope immunoassay, as described in [14]. Plasma and CSF ADNP (Abcam ELISA kit, Paris, France) were assessed using sandwich ELISA (cloud-clone corp ELISA kit, catalog # SEJ674HU, 1/250 for plasma samples and 1/10 for CSF samples). After removal of unbound conjugates, bound enzyme activity was assessed by use of a chromogenic substrate for measurement at 450 nm by a microplate reader (Flex Station 3, Molecular Devices, Ltd., Wokhingham, UK). 

### 2.6. Data Analysis

Statistical analyses were done with the Statview software (Statview 3, Abacus Corporation, Baltimore, MD, USA). We used student *t*-tests and two-way ANOVA followed by the Bonferroni/Dunnet post hoc test. Results are expressed as mean ± SEM (standard error of the mean). Data were considered significant when *p* < 0.05. 

## 3. Results

### 3.1. ADNP Protein Level Is Linked to DYRK1A Protein Level in the Mouse Brain 

There are three independent mouse regions homologous to HSA21, the largest region is found on mouse chromosome 16 (MMU16), which contains the gene encoding DYRK1A. Therefore, we used the Dp(16)1Yey mouse model, which carried the duplication of the entire mouse chromosome 16 (MMU16) region syntenic to HSA21 [23]. ADNP protein level was increased in the brain of Dp(16)1Yey compared to WT mice (Figure 1). To show if the increased ADNP level is due to DYRK1A overexpression, we used a transgenic line overexpressing DYRK1A, the mBACtgDyrk1A mouse model [22]. The level of protein ADNP was also found to be increased in brain of mBACtgDyrk1A mice (127.6 ± 9.7 vs. 100 ± 4, n = 8 for each group, *p* < 0.05). Furthermore, a positive correlation was found between the ADNP and DYRK1A protein level (*p* < 0.02 with a ρ = 0.615). 

### 3.2. DYRK1A Protein Level Is Modified in LCLs from DS Patients with AD Dementia

Recent studies have demonstrated that the protein DYRK1A is truncated in symptomatic AD. Rge C-terminal truncated protein has stronger activity than its full-length form in promoting Tau phosphorylation [26]. This increase in the truncated form is also associated with a decrease in the level of the full-length form [26,27]. Using the anti-Dyrk1A antibody 7D10 targeting the C-terminal region of DYRK1A (Figure 2a), we observed a decrease in the level of full-length form in LCLs from dDS patients compared to level in LCLs from aDS patients (117 ± 23 vs. 227 ± 34) (Figure 2b,c). In turn, this level was increased in LCLs from aDS patients when compared to euploid healthy individuals (227 ± 34 vs. 100 ± 17) (Figure 2b,c). Therefore, we performed Western blot analyses using the D1694 antibody targeting the N-terminal region of the protein DYRK1A (Figure 2a) and observed a decreased level of the full-length form of DYRK1A with an increased level of the truncated form (50 kDa) in LCLs of patients with dDS compared to the level in LCLs from individuals with aDS (317 ± 37 vs. 214 ± 22) (Figure 2b,d), the level of this truncated form being increased in LCLs of patients with aDS compared to the level in LCLs from euploid healthy individuals (214 ± 22 vs. 100 ± 15 (Figure 2d). Unfortunately, it was not possible to detect ADNP in LCLs.

### 3.3. DYRK1A Protein Level Is Modified in CSF and Plasma in DSAD

Table 1 shows the demographic data for the participants included in the study. Using the same antibodies (Figure 2a), we assessed the levels of different forms of DYRK1A in CSF and plasma. Given that the mean age of aDS, pDS, and dDS was 40 years, 52 years, and 55 years, respectively, we used two groups of control patients, with a mean age being 39 years and 54 years. As we found no statistical difference between these two groups on controls, these two groups were treated as separate but as one group designated as controls (Table 1). Furthermore, we found no sex differences for CSF and plasma DYRK1A levels. 

Compared to controls, full-length and truncated forms of DYRK1A as well as ADNP levels were higher in CSF and plasma of aDS individuals (Table 1). 

Compared to asymptomatic group (aDS), the full-length form of DYRK1A decreased in CSF and plasma in the symptomatic groups (pDS and dDS), with no differences between the pDS and dDS groups. Conversely, the truncated form level of DYRK1A was found to be increased in the CSF of pDS and dDS groups compared to aDS, but in plasma, the increase was only statistically significant for the dDS group (Table 1). The ADNP level was decreased in the CSF and plasma of the pDS and dDS groups (Table 1). Compared to the pDS group, ADNP levels decreased in CSF but increased in plasma from the dDS group (Table 1). 

### 3.4. DYRK1A Protein Level Is Modified in CSF and Plasma from Sporadic AD Patients

To show if results are specific for patients with DS, we also used a cohort of AD patients compared to age-matched euploid controls (Table 2). Table 2 shows the demographic data for the participants included in the study, with a mean age of 67 years for controls and 64 years for AD patients. No differences were found between female and male patients in plasma and CSF DYRK1A levels. As previously reported [14,15], the full-length form level of DYRK1A was decreased in plasma of AD patients compared to controls (Table 2). This decrease was also found in the CSF (Table 2). Conversely, the truncated form level of DYRK1A was found to be increased in the CSF and plasma of AD patients, the difference being only statistically significant in CSF. Compared to controls, the ADNP level decreased in CSF but increased in plasma (Table 2). 

## 4. Discussion

Here, we report that the full-length form level of DYRK1A is decreased both in plasma and CSF of symptomatic DS groups (pDS and dDS) compared to the asymptomatic DS group (aDS), and in sporadic AD patients compared to controls. These changes related to symptomatic AD agree with sporadic AD. A similar phenomenon in magnetic resonance spectroscopy was previously found [28]. Conversely, the truncated form level of DYRK1A was found to be increased in CSF and plasma. The proteolytic cleavage of DYRK1A by calpain 1 gives a truncated form with a higher kinase activity towards Tau phosphorylation, this cleavage being positively correlated with the 3R-Tau/4R-Tau ratio in the brains of AD patients [26]. We also observed a decrease in the full-length form of DYRK1A with an increased level of truncated form in LCLs of patients with dDS compared to the level in LCLs of patients with aDS. Neurodegeneration occurs in part driven by neuroinflammation. In this respect, much evidence emphasizes the pivotal role of calpain hyperactivation in neurodegenerative diseases [29]. DYRK1A comprises a signal peptide for protein degradation, named the PEST sequence, which is critical for calpains-dependent proteolytic cleavage [30]. We previously found not only a decrease in the full-length form level of DYRK1A, but also an increased calpain activity in the brain of mBACtgDyrk1A mice after lipopolysaccharide injection, which induces an inflammatory response [31]. Overactivation of calpain 1 is found in the brain of AD patients [32]. Even if DS patients with AD have a distinct neuroinflammatory phenotype compared to sporadic AD [33], we could hypothesize that the decrease of full-length form of DYRK1A associated with the increase of truncated form could be due to an increase of calpain 1 activity in symptomatic DS groups. 

ADNP has been demonstrated to be neuroprotective [34]. Active fragments of ADNP, NAPVSIPQ (NAP), have protective effects in the context of neuroinflammation [35]. They have shown therapeutic potential for developmental delay and learning deficits in a DS model [36,37]. NAP also showed neuroprotective effects in mice models of chronic neurodegeneration such as AD; NAP treatment of transgenic mice initiated at an early stage reduced both Aβ and tau pathology [38]. ADNP-haploinsufficient mice exhibit age-related tauopathy, neurodegeneration, and cognitive deficits [39]. In the transgenic mice that overexpress the mutated tau 4R species in the cerebral cortex, an increase in cortical ADNP has been reported preceding tauopathy [40]. However, with aging, the cortical ADNP decreases [41]. Here, we found an increased ADNP level in the brain of Dp(16)1Yey mouse model, with this increase correlating positively with DYRK1A protein level. Even if ADNP is expressed in many immune system cells [42], we did not detect it in LCLs certainly due to too low protein expression. However, even if we did not confirm the correlation in LCLs, we confirmed the increased ADNP level in the CSF and plasma in the preclinical stage of DS (aDS). Taken together, these results raise the possibility that peripheral ADNP could be used to track disease progression. 

Therefore, we analyzed ADNP levels in the CSF and plasma and found a decreased level in the symptomatic DS groups (pDS and dDS) compared to the asymptomatic DS group (aDS). However, when compared to the pDS group, the ADNP level is decreased in the CSF but increased in the plasma of the dDS group; these results were similar in sporadic AD patients. Malishkevich et al. [21] demonstrated that ADNP plasma/serum levels discriminated well between cognitively normal elderly, MCI, and AD dementia participants. Taken together, these results suggest that the ADNP measures could potentially be a useful surrogate marker for the onset and progression of AD disease in individuals with DS.

## 5. Conclusions

In summary, our present findings demonstrate a decrease in the full-length form of DYRK1A with an increase in the truncated form in plasma and CSF of symptomatic DS groups compared to the asymptomatic DS group, and in patients with sporadic AD compared to controls. Therefore, our results indicate that the altered levels of the full-length form and truncated form of DYRK1A are associated to AD pathology in individuals with DS. Plasma and CSF levels of ADNP were found to increase in the preclinical stage of DS and decreased in the symptomatic groups of DS. A comparison between plasma and CSF for pDS and dDS showed similar results than in sporadic AD patients compared to controls. Therefore, these results demonstrate that altered levels of ADNP are associated to AD progression in individuals with DS. In conclusion, this study shows for the first time that the comparison between full-length and truncated form levels of DYRK1A coupled with ADNP levels could be used in trials targeting pathophysiological mechanisms of dementia in individuals with DS.

## Figures and Tables

**Figure 1 biomedicines-10-01380-f001:**
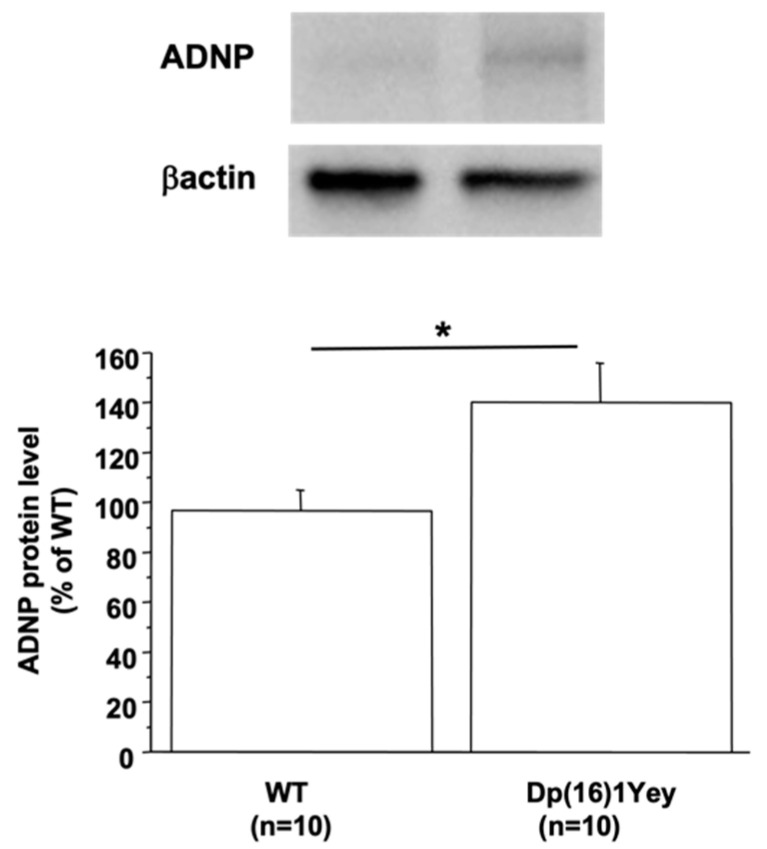
ADNP protein levels in brain of Dp(16)1Yey mice. Western blotting showing ADNP in brain of control (WT) and Dp(16)1Yey mice. Values were obtained by normalization of images from ADNP to βactin (for Western blot) or Ponceau-S coloration (for slot blot). Statistical analysis was done with the Student’s *t*-test using Statview software. Data were normalized to the mean of control (WT) mice. The results are represented as means ± SEM. n = number of mice. * *p* < 0.05.

**Figure 2 biomedicines-10-01380-f002:**
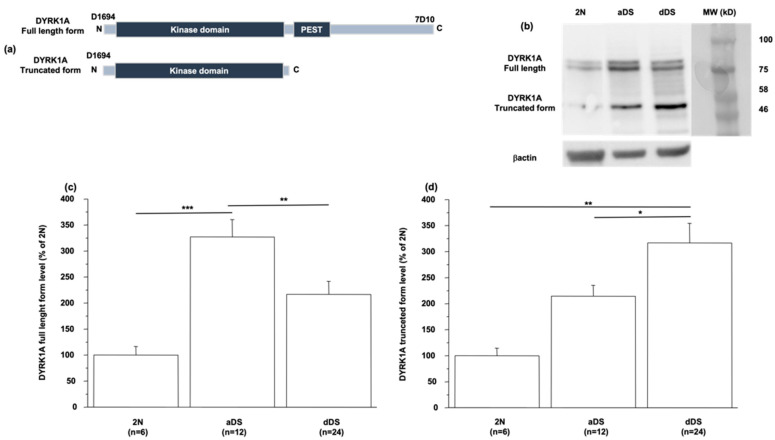
Full-length and truncated DYRK1A levels in LCLs. (**a**) Antibodies used for full-length (D1694) and truncated forms of DYRK1A detection. (**b**) Western blotting showing full-length and truncated form levels of DYRK1A in LCLs from healthy individuals (2N) and DS patients with (dDS) or without (aDS) AD dementia. Proteins were performed by Western blotting and values were obtained by normalization of images to βactin. (**c**,**d**) Data of full-length and truncated forms were normalized to the mean of healthy individuals (2N) and results are represented as mean ± SEM. n = number of LCLs. * *p* < 0.05; ** *p* < 0.005.; *** *p* < 0.001.

**Table 1 biomedicines-10-01380-t001:** Demographics and biomarker data of participants with DS and controls.

	Controls	aDS	pDS	dDS
Mean age at collection (years)	45	40	52	55
N (CSF)	19	45	22	27
Female/male (CSF)	13F/6M	22F/23M	11F/11M	12F/15M
CSF DYRK1A long form (ng/mL)	5 ± 0.3	7.4 ± 0.7 *	4 ± 0.35 ^$$$^	4 ± 0.3 ^$$$^
CSF DYRK1A long and short form (ng/mL)	7.1 ± 0.8	9.4 ± 1 *	14.8 ± 2.2 ^$^	19.6 ± 3.1 ^$$$^
CSF ADNP (ng/mL)	34.9 ± 1.3	40.1 ± 2 *	33.1 ± 2.1 ^$^	26.3 ± 2.5 ^$$$, §^
N (Plasma)	19	77	22	57
Female/male (Plasma)	10F/9M	36F/41M	10F/12M	28F/29M
Plasma DYRK1A long form (ng/mL)	2.8 ± 0.4	3.4 ± 0.2 *	2.1 ± 0.3 ^$$^	2.4 ± 0.2 ^$$^
Plasma DYRK1A long and short form (ng/mL)	3.7± 0.5	5.2 ± 0.35 *	6.5 ± 1	7.65 ± 0.6 ^$$$^
Plasma ADNP (ng/mL)	1.5 ± 0.1	2 ± 0.1 *	0.6 ± 0.1 ^$$$^	1.5 ± 0.1 ^$$, §§^

N: number of samples; aDS: asymptomatic AD; pDS: prodromal AD; dDS: AD dementia; CSF: cerebrospinal fluid; controls: controls euploid healthy controls. Symbols designate significant differences between groups. * *p* < 0.05 (aDS vs. controls); ^$^ *p* < 0.05; ^$$^ *p* < 0.005; ^$$$^ *p* < 0.0001 (pDS and aDS vs. DS); ^§^ *p* < 0.05; ^§§^ *p* < 0.0001 (dDS vs. pDS).

**Table 2 biomedicines-10-01380-t002:** Demographics and biomarker data of participants with AD and controls.

Group	Controls	AD
N	9	18
Mean age at collection (years)	67	64
Female/male	5F/4M	12F/6M
CSF DYRK1A long form (ng/mL)	5.7± 0.4	3.7 ± 0.6 *
CSF DYRK1A long and short form (ng/mL)	9.8 ± 2.4	21 ± 3.6 *
CSF ADNP (ng/mL)	35.5 ± 4.6	24.2 ± 2.2 *
Plasma DYRK1A long form (ng/mL)	2 ± 0.3	1.4 ± 0.1 *
Plasma DYRK1A long and short form (ng/mL)	3.5 ± 0.5	4.6 ± 0.3
Plasma ADNP (ng/mL)	1.4 ± 0.1	2.5 ± 0.2 **

N: number of samples; AD: Alzheimer’s disease; CSF: cerebrospinal fluid; controls: controls euploid healthy controls. Symbols designate significant differences between groups. * *p* < 0.05; ** *p* < 0.0001.

## Data Availability

The data analyzed during the current study are available from the corresponding author on reasonable request.

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
