# Peer review of "DYRK1A and Activity-Dependent Neuroprotective Protein Comparative Diagnosis Interest in Cerebrospinal Fluid and Plasma in the Context of Alzheimer-Related Cognitive Impairment in Down Syndrome Patients"

_biomedicines, 2022, doi:10.3390/biomedicines10061380_

Round 1

Reviewer 1 Report

This is a well-thought out study and well-written manuscript, with the results supporting the conclusions. I have only minimal suggestions. In line 57, I would change the phrase "since 12-13 years of age" to something a little clearer, like "starting at age 12-13 years." Tables 1 and 2 need minor modifications. In table 1, column 1, row 10, one of the "short" words can be deleted; column 2, final row should have the result 2.8 +/- 0.4 centered instead of at the top of the row; final dDS column can have the open parenthesis deleted from before 12F. Table 1 continues on page 7 and it really would be better if it could be on the same page. The result for plasma DYRK1A long and short form for aDS should be cenetered rather than at the top of the row. This same issue is present in Table 2, in rows 8 (column 2) and 9 (columns 2 and 3). Also the F and M for female and male should be capitalized to keep the format consistent with Table 1.

Author Response

This is a well-thought-out study and well-written manuscript, with the results supporting the conclusions. I have only minimal suggestions.

  1. In line 57, I would change the phrase "since 12-13 years of age" to something a little clearer, like "starting at age 12-13 years."

Answer:  We have changed the sentence by: DS brains exhibit Aβ diffuse plaques starting at age 12–13 years, which are observed in the preclinical and prodromal stages of sporadic AD.

  1. Tables 1 and 2 need minor modifications.

In table 1, column 1, row 10, one of the "short" words can be deleted; column 2, final row should have the result 2.8 +/- 0.4 centered instead of at the top of the row; final dDS column can have the open parenthesis deleted from before 12F. Table 1 continues on page 7 and it really would be better if it could be on the same page. The result for plasma DYRK1A long and short form for aDS should be centered rather than at the top of the row.

Answer: we corrected the layout of the Table 1 and of the article to have the table on the same page.

This same issue is present in Table 2, in rows 8 (column 2) and 9 (columns 2 and 3). Also, the F and M for female and male should be capitalized to keep the format consistent with Table 1.

Answer: we corrected the layout of the Table 2.

Reviewer 2 Report

The authors should explain why it was not possible to detect ADNP in LCLs, and what this can hampers in the results.

Minor typo amendments are needed, as follows.

Lines 72-73. Please amend “Core AD biomarkers of AD….” In “Core biomarkers of AD….”

Line 235. Please amend “controls patients” in “control patients”.

Line 301. Please amend “increase” in “increased”.

Author Response

The authors should explain why it was not possible to detect ADNP in LCLs, and what this can hampers in the results.

Answer: we discussed this point in the discussion, line 302 by: Here we found an increased ADNP level in the brain of Dp(16)1Yey mouse model, this increase correlating positively with DYRK1A protein level. Even if ADNP is expressed in many immune system cells [42], we did not detect it in LCLs certainly due to too low protein expression. However, even if we did not confirm the correlation in LCLs, we confirmed the increased ADNP level in the CSF and plasma in the preclinical stage of DS (aDS).

Minor typo amendments are needed, as follows.

Lines 72-73. Please amend “Core AD biomarkers of AD….” In “Core biomarkers of AD….”

Answer: we corrected as mentioned.

Line 235. Please amend “controls patients” in “control patients”.

Answer: we corrected as mentioned.

Line 301. Please amend “increase” in “increased”.

Answer: we corrected as mentioned.